# miR 31-3p Has the Highest Expression in Cesarean Scar Endometriosis

**DOI:** 10.3390/ijms23094660

**Published:** 2022-04-22

**Authors:** Maria Szubert, Anna Nowak-Glück, Daria Domańska-Senderowska, Bożena Szymańska, Piotr Sowa, Aleksander Rycerz, Jacek R. Wilczyński

**Affiliations:** 1Clinic of Surgical and Oncologic Gynecology, 1st Department of Gynecology and Obstetrics, Medical University of Lodz, M. Pirogow’s Teaching Hospital, Wilenska 37 St., 94-029 Lodz, Poland; annanowak27@wp.pl (A.N.-G.); aleksander.rycerz@stud.umed.lodz.pl (A.R.); jrwil@post.pl (J.R.W.); 2Club 35, Polish Society of Gynecologists and Obstetricians, 53-125 Wrocław, Poland; 3Department of Molecular Medicine, Medical University of Lodz, Pomorska 251 St., 92-213 Lodz, Poland; daria.domanska@umed.lodz.pl; 4CoreLab, Central Scientific Laboratory of the Medical University of Lodz, Mazowiecka 6/8 St., 92-215 Lodz, Poland; bozena.szymanska@umed.lodz.pl; 5Department of Pathology, M. Pirogow’s Teaching Hospital, Wilenska 37 St., 94-029 Lodz, Poland; noctee@wp.pl

**Keywords:** miRNA, endometriosis, endometrial cyst, cesarean scar endometriosis, deep infiltrating endometriosis

## Abstract

Micro-RNAs expression can vary between different forms of endometriosis, but data on miRNA expression in cesarean scar endometriosis is lacking. The present study is comprised of 30 patients with endometriosis in the cesarean scar (scar endometriosis, SE), 14 patients with deep infiltrating endometriosis (DIE), 47 patients with endometrioma (ovarian endometrial cyst, OE), and 33 patients with healthy ovarian tissue as the control group (CG). In the initial experiment to identify possible dysregulated miRNAs, the levels of 754 miRNAs in formalin-fixed paraffin-embedded tissue (FFPE) samples from OE, high-grade ovarian cancer, endometrioid ovarian cancer, and CG were measured. We identified seven potentially dysregulated miRNAs: miR-1-3p, miR-31-3p, miR-125b-1-3p, miR-200b-3p, miR-548d, miR-502, and miR-503. We then examined the expression profiles of each of these miRNAs individually in the SE, DIE, OE, and CG FFPE samples using RT-qPCR. miR-31-3p had significantly higher levels of expression and miR-125b-1-3p had significantly lower levels of expression in SE compared to the controls. Overall, the higher expression levels of miR-31-3p and the lower expression levels of miR-125b-1-3p are consistent with the benign nature of SE. Importantly, the results of the present study demonstrate the possibility of using miRNA to monitor the risk of malignant transformation of endometriosis tissue.

## 1. Introduction

Endometriosis is a benign condition, with symptoms connected with the location of the endometriosis foci on the utero-sacral ligament, in the ovary, on the peritoneum in the pouch of Douglas, or in the caesarean scar [1]. It has a negative impact on the quality of life because of persistent cyclic pain [2,3]. Extragenital endometriosis may arise in a cesarean scar due to the iatrogenic implementation of functional endometrial tissue outside of the uterus. This form of endometriosis is called cesarean scar endometriosis and can cause pain and palpable tumors in the abdominal wall. The pathogenesis of endometriosis has been an object of research for many years [4]. In the last decade, the molecular differences between the eutopic endometrium (inside the uterine cavity) and endometriotic lesions have been investigated, and it was found that many mRNA transcripts are differentially regulated in endometriotic lesions compared with eutopic tissue [5]. The discoveries led to the new term for ectopic endometrium—“endometrium-like tissue” [6].

MicroRNAs (miRNA) are single-stranded, noncoding, small RNA molecules (20–40 nucleotides) that are involved in the regulation of gene expression most often by inducing translational repression [7]. Changes in miRNA expression are directly mirrored in mRNA translation. Some miRNAs act in more than one signaling pathway. They control biological pathways involved in proliferation, apoptosis, migration, cell cycle control, differentiation, and angiogenesis [8]. Aberrant functioning of these processes can lead to a variety of pathologies, including malignant transformation. Recently it was proven by Laudanski et al. that eutopic endometrium in endometriosis women has a different miRNA pattern than that from healthy controls [9]. This difference is a likely factor in many of the pathological pathways in endometriosis. For example, miRNA-200 family, which is known to play a crucial role in “epithelial-mesenchymal transition” [10], is dysregulated in endometrioid and clear cell ovarian cancer which represent endometriosis associated ovarian cancer (EAOC) [11,12]. PTEN mutation has a significant role in the malignant transformation of endometriosis tissue, and its down-regulation by a number of miRNAs is reported to occur in endometriosis and ovarian cancer [12]. miR-200a-3p is one of the miRNAs that target PTEN in endometriosis and ovarian cancer [12] and miR-200b targets PTEN in metastatic prostate cancer [13]. Recently, it has been shown that miRNAs are exceptionally stable and can be readily and reliably detected in most tissues, and in formalin-fixed paraffin-embedded tissue (FFPE) samples [14]. Hanna J et al. published a review on the clinical utility of miRNAs—both as indicators for disease progression in patients receiving oncological treatment as well as therapeutic agents, mainly in oncology [15].

In the present study, we compared miRNA expression in different types of endometriosis tissues. This is the first report of miRNA expression in cesarean scar endometriosis. Data from this study adds to basic knowledge in understanding the relationships between miRNAs and endometriosis types. Our findings also suggest that miRNA expression profiles can be used to monitor the risk of malignant transformation of endometriosis.

## 2. Results

The expression levels of 754 human miRNA genes were assessed in a screening cohort of randomly chosen FFPE endometriosis samples (OE) (*n* = 20) and controls (CG) (*n* = 20) and ovarian cancer samples (*n* = 20 + 20) using a MicroRNA array: the Heat Map is shown in Appendix A. hsa-miR-191-5p (assay ID: 477952_mir) served as the reference gene (calibrator). hsa-miR-191-5p was chosen because the miR-191 precursor is highly expressed but does not show tissue specificity [16,17], it is not reported to be dysregulated in ovarian endometriosis [5] or EAOC [12], and a search of pubmed did not reveal any studies reporting an association of hsa-miR-191-5p with endometriosis. miRNAs with altered expression profiles, a 2 fold change in their relative quantification (RQ) value compared to the reference hsa-miR-191-5p, were chosen for inclusion in the present study (Table 1): our analysis of miRNA expression in ovarian cancer will be published in a future report. In addition, miR-1-3p was included in the present study based on its upregulation (1.4-fold) taken in conjunction with other studies reporting its upregulation in endometriosis [18].

Chosen miRNAs were as follows (detailed description in Section 4):Name in DataBase Accession Number hsa-miR-125b-1-3pMIMAT0004592hsa-miR-31-3pMIMAT0004504hsa-miR-1-3pMIMAT0000123hsa-miR-191-5pMIMAT0000440hsa-miR-200b-3pMIMAT0000318 hsa-miR-502-5pMIMAT0002873hsa-miR-503-5pMIMAT0002874hsa-miR-548d-5pMIMAT0004812

Expression of the seven selected miRNAs in control and three different forms of endometriosis tissue was assessed by RT-qPCR with hsa-miR-191-5p (RQ = 1) as the calibrator. The Box and Whisker profiles of the selected miRNAs are shown in Figure 1: extreme outliers are not shown in Figure 1. The means and medians are shown in Table 2. All of the data, including extreme outliers is shown in Appendix A.

miR-31-3p had significantly higher expression in SE compared to the controls (*p* = 0.001). miR-31-3p also had significantly higher expression in SE compared to OE (*p* = 0.0004). While not statistically significant, miR-31-3p was expressed at higher levels in SE than in DIE (Figure 1). In addition, the median levels of miR-31-3p expression in all three endometriosis tissues was markedly higher than the control group (Table 2).

miR-125b-1-3p had significantly lower levels of expression in SE compared to the controls (*p* = 0.0001). The highest levels of miR-125b-1-3p were in the DIE samples and expression in DIE was significantly higher than in OE (*p* = 0.0436). The median levels of expression of miR-125b-1-3p was lower in all three endometriosis tissues than the control group, with expression in the control group at least 2-fold higher than in the endometriosis tissues (Table 2). miR-200b-3p was expressed at higher levels in DIE compared to the controls (*p* = 0.025). In addition, the median levels of expression of miR-200b was much higher in all three endometriosis tissues than in the control group (Table 2). miR-1-3p was expressed at higher levels in DIE than in OE (*p* = 0.001). In all tissues examined, miR-502 was expressed at exceedigly high levels compared to expression of the other miRNAs examined. miR-502 expression was significantly higher in DIE compared to OE (*p* = 0.018), however, the median level of miR-502 expression was lower in DIE than in OE. miR-503 and miR-548d were not expressed at significantly different levels in any of the groups. The median levels of expression of miR-1-3p, miR-502, miR-503, and miR-548 were all within 2-fold of the control group (Table 2).

### Correlations

Correlation graphs are shown in Appendix A. In SE samples there was a positive correlation between miR125b-1-3p and miR-548d (R = 0.484; *p* = 0.02) (Figure 2), between miR-125b-1-3p and miR-31-3p (R = 0.417; *p* = 0.043), and between miR-200b-3p and miR-31-3p (R = 0.380; *p* = 0.01). In DIE samples there was a positive correlation between miR-1-3p and miR-125b-1-3p (R = 0.738; *p* = 0.037). In OE samples there was a positive correlation between miR-502 and miR-503 (R = 0.402; *p* = 0.005) and between miR-502 and miR-548d (R = 0.527; *p* = 0.001).

## 3. Discussion

This is the first study to explore iatrogenic form of endometriosis—endometriosis in the cesarean scar. miRNA expression in several forms of endometriosis has already been reported [19], however, this is the first report of miRNA expression in SE. Of the miRNAs described in the literature, nearly 1000 of them are accessible in manufactured sets. Many miRNAs are poorly known, and their roles as non-coding factors need to be established. In the present study, miR-31-3p and miR-125b-1-3p were expressed at significantly different levels in SE compared to the controls: miR-31-3p was higher in SE than in the controls and miR-125b-1-3p was lower in SE than in the controls. In addition, the median expression levels of miR-31-3p were higher in SE than in the other groups and median levels of miR-125b-1-3p were lower than in the other groups. Importantly, the median levels of miR-31-3p were higher in all 3 types of endometriosis than in the controls and the median levels of miR-125b-1-3p were lower in all 3 types of endometriosis than in the controls.

It has been many years since the initial report on miRNA-31-3p [20]. This miRNA was later described in the Xenopus frog model organism database [21] and then in several studies on colorectal, breast cancer, mesothelioma, and thyroid cancer [22,23]. These studies have reported association of miR-31 expression with both tumor-suppressive and tumor-promoting activity. An early in vitro study using vascular smooth muscle cells from male Sprague-Dawley rats showed that up-regulation of miR-31 by the MAPK/ERK pathway resulted in downregulation of Large Tumor Suppressor Homolog 2 (LATS2) and enhanced cellular proliferation [24]. miR-31 also targets negative regulators of Ras signaling, resulting in a positive feedback loop for RAS/MAPK/ERK signaling [25]. However, most cancers show down-regulation of miR-31 expression. For example, in thyroid cancer, down-regulation of miR-31 was responsible for malignant progression of papillary thyroid carcinoma cells [22] and down-regulation of miR-31 was associated with a worse clinical outcome for mesothelioma and hepatocellular carcinoma patients [23,26].

One explanation for these conflicting findings is that in many cancers Ras carries a mutation that prevents binding of negative regulators. In such cancers, inhibition of negative regulators by miR-31 would not have an effect, allowing the anti-tumorigenic properties of miR-31 to predominate. In our study, however, the relatively high expression of miR-31-3p in all three types of endometriosis tissues suggests the possibility of a pathway that represses miR-31 activation of RAS/MAPK/ERK signaling in these benign lesions, and that the overall activity of miR-31 in these tissues is tumor-suppressive.

miR-31 also down-regulates the drug transporter ABCB9, inhibiting cisplatin-induced apoptosis [26]. Importantly, low expression of ABCB9 is associated with poor survival of ovarian cancer patients [27]. These reports suggest that up-regulation of miR-31 and accompanying down-regulation of ABCB9 would be associated with ovarian cancer, however, our findings indicate that miR-31-3p is up-regulated in SE, a benign lesionThus, similarly to the RAS/MAPK/ERK signaling pathway discussed above, these findings suggest the presence of opposing pathways associated with miR-31-3p. Future studies investigating the activities of pathways in endometriosis tissues that are activated by up-regulation of miR-31 in cancer tissues need to be carried out. 

While miR-31-3p expression was found to be correlated with expression of miR125b-1-3p in scar endometriosis tissue, expression of miR-125b-1-3p was significantly down-regulated in scar endometriosis tissue compared to healthy tissue. However, like miR-31, expression of miR-125b is associated with both anti-tumorigenic and pro-tumorigenic activities [28,29]. Up-regulation of miR-125b is reported to to suppress proliferation of bladder urothelium by down-regulating E2F3 and inhibiting E2F3-Cyclin A2 signaling and transition through G1/S [30], inhibit proliferation of breast cancer cells by acting in concert with miR-125a and miR-205 to down-regulate the epidermal growth factor receptors erbB2 and erbB3 [31] and to suppress endometrial cancer invasion by down-regulating expression of erbB2 [32,33]. On the other hand, up-regulation of miR-125b is reported to have pro-tumorigenic activity in endometrial cells by targeting tumor protein p53 inducible nuclear protein 1 [34]. Notably, while the expression of miR-125b-1-3p was significantly lower than the controls only in scar endometriosis tissue, the median levels of expression of miR-125b-1-3p was lower than the controls in all three endometriosis tissue types (Table 2). This suggests that in these tissues, up-regulation of miR-125b-1-3p may be associated with pro-tumorigenic activity. The premise that up-regulation of miR-125b-1-3p may be associated with pro-tumorigenic activity is in agreement with the finding that miR-125b targets tumor protein p53 inducible nuclear protein 1: before tumorigenic transformation is able to occur, pathways inhibiting aberrant proliferation and pathways that lead to the death of cells that have undergone aberrant proliferation, especially p53-associated pathways, must be suppressed [34]. Consequently, in tissues in which the p53 pathway is suppressed, such as tumor tissue, miR-125b would suppress proliferation by down-regulating erbB2/3 and E2F3, however, in tissues with intact p53 pathways, such as endometriosis tissue, miR-125b suppression of p53 pathways would be pro-tumorigenic. 

The other miRNA that was statistically different from the control was miR-200b-3p. The median levels of miR-200b-3p were higher in all three endometriosis tissue types than in the controls and the levels in DIE were significantly higher than in the controls. miR-200-3p targets a multitude of genes and, similarly to miR-31 and miR-125b, miR-200b-3p is reported to have both anti-tumorigenic and pro-tumorigenic activities [35]. While miR-200b-3p appears to have anti-tumorigenic activity in most tissues, high expression of miR-200b is associated with ovarian cancer progression and increased tumor stage [36,37]. This suggests that in endometriosis tissue, miR-200b-3p may be associated with pro-tumorigenic pathways. Among other pathways targeted by miR-200b, up-regulation of miR-200b is reported to target pathways associated with cellular proliferation [38]. Our findings of increased median expression levels of miR-200b in all three endometriosis tissue types, coupled with the pro-proliferation activity of miR-200b, is in agreement with the fact that endometriosis is a proliferative disorder.

Elucidating the function of miRNAs in a particular tissue is complicated by their multiple functions in different tissues and under different conditions. However, our findings support the known fact that endometriosis is a benign proliferative disorder. The lower median level of expression of miR-125b-1-3p in all three types of endometriosis tissue types is consistent with (1) the finding that up-regulation of miR-125b suppresses tumor protein p53 inducible nuclear protein 1 and (2) the fact that endometriosis is benign. The higher median levels of expression of miR-200b-3p in all three types of endometriosis tissue types is consistent with (1) the reports that miR-200b down-regulates cell proliferation regulatory/inhibitory pathways and (2) the fact that endometriosis is a proliferative disorder.

Similarly to miR-125b and miR-200b, miR-31-3p is associated with both anti-tumorigenic and pro-tumorigenic activities. The higher mean and median levels of expression of miR-31-3p in all three types of endometriosis tissue suggest that miR-31-3p has tumor-suppressive activity in endometriosis tissues.

Ovarian endometriosis (OE) is a known risk factor for the development of ovarian cancer [39,40]. Cesarean scar endometriosis (SE) is thought to be less likely than OE to evolve into an atypical or malignant lesion. miR-31-3p and miR-125b-1-3p may play a role in the lower susceptibility of SE to undergo malignant transformation. The median expression level of miR-31-3p is higher in SE than in DIE and the expression of mR-31-3p is significantly higher in SE compared to OE, and both the mean and median levels of expression of miR-125b-1-3p are lower in SE than in OE or DIE. This is in agreement with the premise that SE has low susceptibility to malignant transformation. Importantly, this also suggests that the levels of expression of specific mRNAs may be used to monitor the risk of malignant transformation of endometriosis tissue.

The most facile, and easiest on the patient, method of obtaining samples for determining miRNA expression levels is from bodily fluids. Misir et al. examined the levels of miR-200c in the blood of endometriosis patients, and, similar to our results with miR-200b-3p, reported that patients with endometriosis had higher serum levels of miR-200c than the control group [41]. However, Rekker et al. reported that the circulating levels of miR-141, miR-200a, and miR-200b vary with blood collection time of day [42]. Thus, great care must be taken when assessing blood levels of miRNAs, and blood levels of miRNAs should include the levels at different times. Rekker et al. also reported that there were differences when miRNA levels were compared between endometriosis patients and patients confirmed by the use of laparoscopy to be endometriosis free and when endometriosis patients were compared with self-reported healthy women. This highlights another confounding factor: serum levels of miRNAs are not specific for any one tissue and will be influenced by expression of the tested miRNA in non-target tissues. Another study reported the expression of miR-200b-3p in menstrual blood-derived mesenchymal stem cells [43]. This study, in agreement with our findings, reported higher expression of miR-200b-3p in women with endometriosis.

## 4. Materials and Methods

This case-control study comprised patients hospitalized and diagnosed for endometriosis between the years 2015–2018 at the Department of Surgical and Oncologic Gynecology, Medical University of Lodz, Poland, for whom histopathological FFPE blocks were identified and key retrospective data were possible to obtain. FFPE was reassessed by a pathologist to choose the right tissue for miRNA analysis. 

Exclusion criteria were as follows: known hormonal therapy before surgery (patient should be at least 3 months without hormonal treatment). We recruited 30 patients with endometriosis in the cesarean scar (scar endometriosis, SE), 14 patients with DIE (deep infiltrating endometriosis), 47 patients with endometrioma (ovarian endometrial cyst, OE), as and as the control group 33 patients with healthy ovarian tissue (CG; ovaries removed with the uterus due to benign conditions, such as large myomas or abnormal uterine bleeding).The mean age of endometriosis patients was 36.6 years, control patients: 58.1 (*p* < 0.01) years. There was no statistical difference between groups in body mass index (BMI), number of pregnancies, deliveries, miscarriages, and comorbidities—detailed characteristics are in Appendix A. The study was approved by the Bioethics Committee of MU of Lodzand, and the study number is RNN 403/18/KE.

### 4.1. Tissue Preparation

Archival FFPE (formalin-fixed, paraffin-embedded) tissues were obtained according to pathological results and after checking for exclusion criteria. A precise identification of affected areas was performed by an experienced pathologist. OE, SE, DIE, and CG tissues were selected, and then carefully dissected on a semiautomatic microtom in order to minimize the risk of contamination with nonaffected tissues. The reference group consisted of normal ovarian tissue that were retrieved after a hysterectomy for benign diseases other than endometriosis or adenomyosis. Ovarian tissue was selected as the control group because of the fact that miRNA expression is tissue-specific (https://ccb-web.cs.uni-saarland.de/tissueatlas, accessed on 6 October 2021) [44] and one of the studied groups was the ovarian endometriosis group, so we deemed the healthy ovary as the most appropriate control tissue. There is discussion about changes in miRNA expression in eutopic endometrium of women with endometriosis and some question regarding external endometriosis, such as SE and DIE, about classifying this tissue as “endometrium-like” and not endometrium itself, making the choice of ovarian tissue as the control less controversial.

### 4.2. RNA Extraction from FFPE Samples

Total RNA was isolated from 1–2 tissue slices (10–20 μm thick) from archival formalin-fixed paraffin embedded (FFPE) blocks using the High Pure FFPET Isolation Kit (Roche, Roche Diagnostics Deutschland GmbH, Mannheim, Germany). Tissue sections were deparaffinized with 100% xylene, washed with 100% ethanol, and dried for 10 min at 55 °C. Protease digestion was performed at 55 °C overnight. Subsequent steps of RNA purification were performed according to the manufacturer’s instruction. Briefly, 325 µL of Binding Buffer and 325 µL of Binding Enhancer was added and the mixture was applied to the columns, washed two times with Wash Buffer and eluted with 50 µL Elution Buffer. The yield and quality of the RNA products were determined using PicoDrop spectrophotometer (Picodrop Limited, Hinxton, UK). The purified total RNA was immediately used for cDNA synthesis or stored at −80 °C until use.

### 4.3. Expression Profile of miRNA Genes

miRNA expression was first screened using TaqMan^®^ Human MicroRNA Array A and B (Applied Biosystems, Foster City, CA, USA). Megaplex™RT Primers Human Pool A and B were purchased from Applied Biosystems and used to prepare the reverse transcription reaction according to manufacturer’s instructions. Amplifications were performed on a 7900HT Fast Real-Time PCR System (Applied Biosystems, Foster City, CA, USA) under the following conditions: 2 min at 50 °C, 10 min at 94.5 °C, and 40 cycles each for 30 s at 97 °C and 1 min at 57 °C. The expression levels of 754 human miRNA genes were assessed in the first step in the screening cohort of randomly chosen FFPE endometriosis samples (OE) (*n* = 20) and controls (CG) (*n* = 20) and in ovarian cancer samples (*n* = 20 for high-grade ovarian cancer + 20 for endometrioid ovarian cancer). miRNAs with altered expression profiles (based on 2-fold change, PCR amplification curves, and confirmed in the literature) were chosen for further investigations in the examined groups of patients: DIE, SE, OE, and CG.

### 4.4. RT-qPCR with Individual cDNA

Individual, quantitative RT-qPCR study was performed using the TaqMan Advanced miRNA Reverse Transcription Kit (Thermo Fisher, Pleasanton, CA, USA) following the manufacturer’s protocol. Briefly, 10 ng of total RNA was used in poly(A) tailing reaction, followed by adapter ligation and reverse transcription. The diluted RT product was preamplified in order to uniformly increase the amount of cDNA for all miRNAs (miR-Amp reaction) and directly used for qPCR or stored at −20 °C.

### 4.5. MicroRNA Assays

Quantification of selected miRNA was done using TaqMan Advanced MicroRNA Assays (Applied Biosystems): hsa-miR-1-3p (Assay ID: 477820_mir), hsa-miR-31-3p (Assay ID: 478012_mir), hsa-miR-125b-1-3p (Assay ID: 478665_mir), hsa-miR-200b-3p (Assay ID: 477963_mir), hsa-miR548d (Assay ID: 480870_mir), hsa-miR-502-5p (Assay ID: 47954_mir), has-miR-503-5p (Assay ID: 47143_mir), and hsa-miR-191-5p (Assay ID: 477952_mir) as a reference gene. The 10 μL qPCR reaction mixture included 2.5 μL of diluted RT product, 5 μL of TaqMan Fast Advanced PCR Master Mix (Applied Biosystems), and 0.5 μL of TaqMan miRNA Assay (20×). The reactions were performed in a 96-well plate at 95 °C for 10 min, followed by 40 cycles of 95 °C for 5 s and 60 °C for 20 s in duplicates. Relative quantification of mRNA was determined by comparative Ct. The miRNA level was calculated as 2−ΔΔCt, while relative expression analysis of the examined gene was presented as an n-fold change in gene expression normalized to a reference gene relative to the control.

### 4.6. Statistical Analysis

Normality was tested using the Shapiro–Wilk test. The significance of differences was analyzed at the level of miRNAs using non-parametric tests (the U Mann–Whitney test and the Kruskal–Wallis test) as the data was not normally distributed. The R-Spearman test was used to assess the correlations between variables. ROC curves were created and the area under the curve (AUC) was calculated in order to evaluate the specificity and sensitivity of the selected miRNAs. Statistical significance was *p* < 0.05. Statistica 13.3 software was used for statistical analysis of obtained data.

## 5. Conclusions

Scar endometriosis had statistically higher expression of miRNA31-3p and statistically lower expression of miR-125b-1-3p compared to the control. Deep infiltrating endometriosis (DIE) and ovarian endometrial cyst (OE) also tended to have higher expression of miRNA31-3p and lower expression of miR-125b-1-3p compared to the control. miR-200b-3p was expressed at a statistically higher levels in DIE compared to the controls, and the median level of expression of miR-200b was higher in all three endometriosis tissues than in the control group. Low expression of miR-125b-1-3p in all three endometriosis tissue types is in agreement with the generally benign nature of endometriosis. Relatively high expression of miR-200b-3p in all three endometriosis tissue types is in agreement with the fact that endometriosis is a proliferative disorder. Relatively high expression of miR-31-3p in all three endometriosis tissue types is in agreement with the fact that endometriosis is a proliferative disorder, however, it also suggests that miR-31-3p may have tumor suppressive activity in endometriosis tissues. Our study supports the possibility of using miRNA expression to monitor of endometriosis tissue.

## Figures and Tables

**Figure 1 ijms-23-04660-f001:**
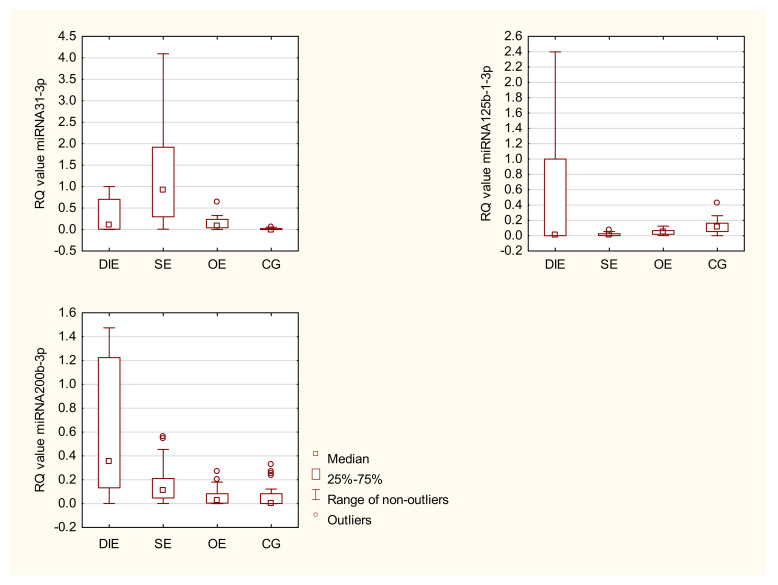
Box and Whisker plots of the Relative Quantification values for miR-31-3p, miR125b-1-3p, miR200b-3p, miR1-3p, miR-502, and miR548d. Extreme outliers are not shown in this figure. All data is presented in Appendix A.

**Figure 2 ijms-23-04660-f002:**
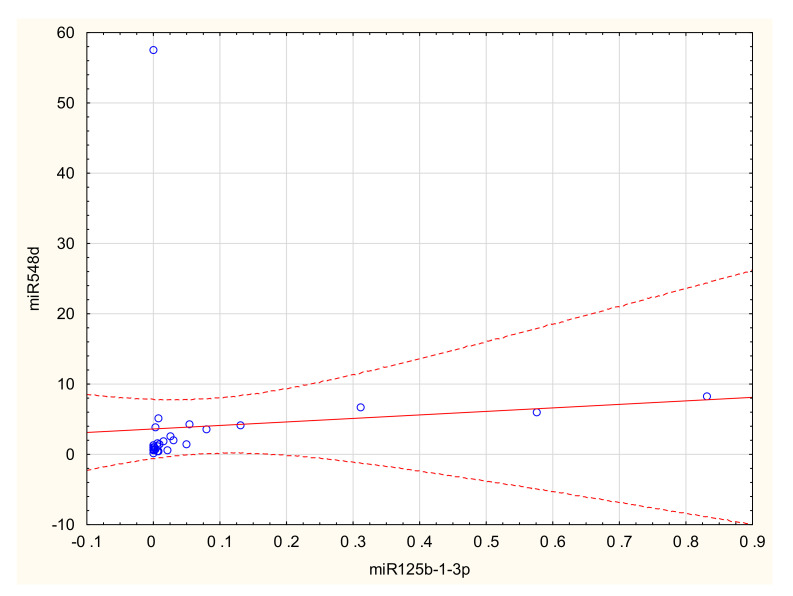
Correlation between miR125b-1-3p vs miR548d in scar endometriosis (R = 0.484; *p* = 0.02) (see also Appendix A).

**Table 1 ijms-23-04660-t001:** Names, assays IDs, and relative expression levels of the miRNAs studied. *n* = 20 for each group; values shown as RQ.

	Groups	Control Group	Ovarian Endometrial Cyst	Endometroid Ovarian Cancer	Serous HG Ovarian Cancer
miRNA					
miR-1-3p Assay ID: 477820_mir		1	1.4029	1.0037	1.0615
miR-31-3p Assay ID: 478012_mir		1	0.9693	8.6679	0.9646
miR-125b-1-3p Assay ID: 478665_mir		1	1.2257	0.1475	0.2446
miR-200b-3p Assay ID: 477963_mir		1	0.2539	15.0699	7.9167
miR-548d Assay ID: 480870_mir		1	1.5972	2.1021	0.8198
miR-502-5p Assay ID: 47954_mir		1	0.1191	0.0055	0.1477
miR-503-5p Assay ID: 47143_mir		1	0.005	0.1759	0.5978

**Table 2 ijms-23-04660-t002:** Relative quantification of the miRNAs in the tissue samples from the different forms of endometriosis and the control group. miR-191-5p (RQ = 1) served as the calibrator.

	miRNA	miR31-3p	miR125b-1-3p	miR200b-3p	miR1-3p	miR502	miR503	miR548d
Studied Cohort								
Scar Endometriosis	Mean SD Median	1.418 ± 1.570 0.961	0.073 ± 0.185 0.007	0.228 ± 0.301 0.113	2.925 ± 13.272 0.0004	2154.744 ± 5663.032 642.001	1.648 ± 0.916 1.768	3.983 ± 10.314 1.269
Control Group	Mean SD Median	0.061 ± 0.176 0.001	0.123 ± 0.96 0.113	0.079 ± 0.158 0.001	0.03 ± 0.002 0.003	1142.603 ± 518.302 1014.267	0.075 ± 0.404 0.586	0.923 ± 0.431 0.832
Ovarian Endometrial Cyst	Mean SD Median	0.176 ± 0.27 0.091	0.202 ± 0.86 0.046	0.06 ± 0.1 0.024	0.003 ± 0.01 0.002	3330.395 ± 13383.09 880.406	1.863 ± 2.16 1.1097	2.251 ± 6.61 0.869
Deep Infiltrating Endometriosis	Mean SD Median	1.9 ± 5.44 0.099	1.434 ± 4.03 0.014	6.284 ± 20.3 0.357	1.069 ± 3.12 0.007	5347.585 ±13,997.62 369.658	2.058 ± 2.54 0.964	3900.219 ± 14,460.058 0.728
**miRNA**	**DIE**	**SE**	**OE**	**CG**	**DIE**	**SE**	**OE**	**CG**
	**min**	**max**	**min**	**max**	**min**	**max**	**min**	**max**
miR31-3p	0.00060	20.52	0.00910	7.68	0.00010	1.59	0.0000	0.964
miR125b-1-3p	0.00010	15.24	0.00010	0.83	0.00120	5.76	0.0003	0.429
miR200b-3p	0.00120	76.62	0.00130	1.44	0.00000	0.58	0.0001	0.691
miR1-3p	0.00000	11.73	0.00000	72.94	0.00010	0.03	0.0002	0.009
miR502	0.86420	52,830.96	0.48730	31,213.67	24.19780	91,947.80	311.9996	2351.290
miR503	0.01250	7.49	0.05290	3.55	0.01420	12.59	0.1399	1.742
miR548d	0.01170	54140.58	0.03510	57.46	0.25490	44.36	0.0096	1.993

## Data Availability

The data presented in this study are available on request from the corresponding author.

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
