# Peer review of "miR 31-3p Has the Highest Expression in Cesarean Scar Endometriosis"

_ijms, 2022, doi:10.3390/ijms23094660_

Round 1

Reviewer 1 Report

Szubert et al studied miRNAs expression in formalin-fixed paraffin-embedded tissue samples in 754 genes. They found that miR 31-3p has the highest expression in cesarean scar endometriosis. The study has some interest with analyses of precious samples with endometrioma,deep infiltrating endometriosis and scar endometriosis.

However, the introduction and the scope of the study has limitations. The description and presentation of data is insufficient. The authors should present the data with PCA, heatmap plots etc. The study has multiple caveats in data analysis and presentation.

  1. Szubert et al mention that only miR-31-3p and miR125b-1-3p had significantly higher expression levels in SE as compared to other form of endometriosis and healthy controls, but they do not mention the level of differences (fold change).
  2. It is uncertain and misleading how “This fact can be used in the future when planning therapy for endometriosis or when establishing the risk of malignant transformation of this disease”.
  3. The figure 1 does not reflect the statement “The highest expression level for every form of endometriosis compared to other 78 miRNAs was observed for miR502 and there was significant difference between endometriosis and healthy ovarian tissue. The same pattern of expression was observed for miR503 although its expression was relatively lower than miR502”
  4. Figure 1 is completely misleading-wrong scale in y axis
  5. Line 57, need to correct the typing error “eutopic”.
  6. Figure 3: what is the p=value and R2?

Author Response

Dear Reviewer,

Thank you very much for your revision. We checked carefully our graphs and statistic according to your suggestion. We implemented heat map as suppl 1. Please see details below – answers are bolded in text:

  1. Szubert et al mention that only miR-31-3p and miR125b-1-3p had significantly higher expression levels in SE as compared to other form of endometriosis and healthy controls, but they do not mention the level of differences (fold change).

Answer: We corrected statement about miR125b-1-3p according to statistical data. We also implemented p value into text under figures and in appropriate places in Results section.

  1. It is uncertain and misleading how “This fact can be used in the future when planning therapy for endometriosis or when establishing the risk of malignant transformation of this disease”.

Answer: Conclusions were written more precisely – see lines 266-270.

  1. The figure 1 does not reflect the statement “The highest expression level for every form of endometriosis compared to other 78 miRNAs was observed for miR502 and there was significant difference between endometriosis and healthy ovarian tissue. The same pattern of expression was observed for miR503 although its expression was relatively lower than miR502”

Answer:

Figure no 1 was corrected, detailed explanation was written in figure legend. The mistake was because figure 1 was first created from the expression levels from the first part of the study (pooled samples). Now it shows mean values of all studied samples grouped accordingly in OE, SE, DIE and CG.

  1. Figure 1 is completely misleading-wrong scale in y axis

j.w.

Answer: Y axis was drawn logarithmic (Excel function) to ensure that all mean values are visible on the scale.

  1. Line 57, need to correct the typing error “eutopic”. – It is not mistake; should be “eutopic” what means “in the cavum uteri, on site”
  2. Figure 3: what is the p=value and R2?

Answer: The information was added to the description of Fig 3 and cited below: Spearman correlation rang

Reviewer 2 Report

 The authors measured miRNAs expression in formalin-fixed paraffin-embedded tissue samples of endometriosis {scar endometriosis (SE), OE, DIE, CG}.

They found that miR-31-3p had significantly higher expression levels in SE as compared to other form of endometriosis and healthy controls.

This is the first report of miRNA expression in cesarean scar endometriosis. The result of this study is interesting for gynecologists and pathologists and can be used in the feature when planning therapy for endometriosis or when establishing the risk of malignant transformation of endometriosis.

   It might be improved on the following issues.

#1. In Figure 1: relative quantification of miRNA, the highest expression level for DIE (deep infiltrating endometriosis); control group was observed for miR548d. Is this a significant difference?

#2. Result, P.5, line 86, “miR125 had significant higher expression in SE comparing to CG”.: According to the figure 1 and 2, this(miR125) seems to indicate a low expression.

#3, Result, P.5, line 88, “miR31-3p had the highest level in SE”: In figure 1, miR31-3p had higher expression level for DIE than SE.

#4. Discussion, P.6. line 112 and line129, ”high expression of miR31 and miR125” same as #2 and #3: However, in figure 1, miR125 seems to indicate a low expression.

#5. Material and Method, RNA extraction from FFPE samples, P.7, line 161,

“Tissue section”: How did the authors get the lesion of endometriosis (endometrial tissue) from the tissue section, such as microdissection?

#6. Are there any cases of malignant transformation in this case control study?

#7. Materials and Methods, P.6, line 154, Why did the authors choose a “healthy ovarian tissue” as a “control group”? Normal endometrial tissue will be suitable.

Author Response

Dear Reviewer,

thank you for all your suggestions that improved our manuscript. We implemented them accordingly:

#1. In Figure 1: relative quantification of miRNA, the highest expression level for DIE (deep infiltrating endometriosis); control group was observed for miR548d. Is this a significant difference?

Answer: no, p was > 0,05

#2. Result, P.5, line 86, “miR125 had significant higher expression in SE comparing to CG”.: According to the figure 1 and 2, this(miR125) seems to indicate a low expression.

Answer: You are right, thank you. This was corrected both in lines 121-126 and 261.

#3, Result, P.5, line 88, “miR31-3p had the highest level in SE”: In figure 1, miR31-3p had higher expression level for DIE than SE.

Answer: Figure 1 was corrected. The statement is true. As we answered for review number 1: Figure no 1 was corrected, detailed explanation was written in figure legend. The mistake was because figure 1 was first created from the expression levels from the first part of the study (pooled samples). Now it shows mean values of all studied samples grouped accordingly in OE, SE, DIE and CG.

#4. Discussion, P.6. line 112 and line129, ”high expression of miR31 and miR125” same as #2 and #3: However, in figure 1, miR125 seems to indicate a low expression.

Answer: Disucssion adjusted as we explained above.

#5. Material and Method, RNA extraction from FFPE samples, P.7, line 161,

“Tissue section”: How did the authors get the lesion of endometriosis (endometrial tissue) from the tissue section, such as microdissection?

Answer: New subheading added – 4.1 with the metodhology of tissue sampling and dissection.

#6. Are there any cases of malignant transformation in this case control study?

Answer: There were no malignant transformation cases.

#7. Materials and Methods, P.6, line 154, Why did the authors choose a “healthy ovarian tissue” as a “control group”? Normal endometrial tissue will be suitable.

Answer: We described it precisely - See new chapter 4.1. Ovarian tissue has been choosen as control group due to the fact that miRNA expression is tissue specific. One of the studied group was ovarian endometriosis group so healthy ovary seemed the most appropriate control tissue. There is also a “big battle” between scientists about changes in miRNA expression in eutopic endometrium of women with endometriosis and discussion about external endometriosis – like SE and DIE, that this tissue is only “endometrium like” and not endometrium itself. 

Reviewer 3 Report

In manuscript “miR 31-3p has the highest expression in cesarean scar endometriosis”, the authors presented results from their retrospective study on the expression of miRNAs in different types of endometriosis. The research topic is novel and the data is interesting. However, the study is not comprehensive enough to come into solid, clinically meaningful conclusion.

The authors used a set of commercial miRNA arrays to screen for miRNA species which are differentially expressed in different types of endometriosis, but it was not detailed in the manuscript. How were 10 patients/group selected? Was there any significant difference in their clinicophysiological parameters? In the array what was used as controls, how was normalization performed?

Out of the 754 miRNAs, how was statistical analysis performed to identify the final 7?

What is the difference between Figure 1 and Figure 2? In Figure 1, miR502 and miR503 levels are roughly in a same range, however, in Figure 2, miR502 level is 1000 times higher? The results are confusing.

The results show some miRNA species are higher in certain types of endometriosis, what do they mean? Can they be linked with pathogenesis of endometriosis?

Minor issue:

In Discussion, the authors wrote “miR125-1-3b is a member of conservative family of proteins”. This is incorrect.

Author Response

Dear Reviewer,

thank you for all your suggestions.

According to your indications we corrected Results section and also M&M section – which is next to last section. Detailed explanation we would also copied below:

In manuscript “miR 31-3p has the highest expression in cesarean scar endometriosis”, the authors presented results from their retrospective study on the expression of miRNAs in different types of endometriosis. The research topic is novel and the data is interesting. However, the study is not comprehensive enough to come into solid, clinically meaningful conclusion.

The authors used a set of commercial miRNA arrays to screen for miRNA species which are differentially expressed in different types of endometriosis, but it was not detailed in the manuscript. How were 10 patients/group selected? Was there any significant difference in their clinicophysiological parameters? In the array what was used as controls, how was normalization performed?

Out of the 754 miRNAs, how was statistical analysis performed to identify the final 7?

Answer:

7 miRNAs with altered expression profiles were chosen out of 754 human miRNA genes. hsa-miR-191-5p (assay ID: 477952_mir) served as a reference gene https://mirbase.org/cgi-bin/mirna_entry.pl?acc=MI0000465 [11]. The expression levels of 754 human miRNA genes were assessed in the first step in the screening cohort of randomly choosen FFPE endometriosis samples (OE) (n=10) and controls (CG) (n=10) and in ovarian cancer group. We tested also cohort of ovarian cancer for further investigations but the results are not included in this manuscript and will not be described here. MiRNAs with altered expression profiles (at least 2 fold change compared to reference hsa-miR-191-5p) were chosen for second step of the experiment after analysis of their amplification curves and analysis of the literature backgrounds. Heat map of the first step of the experiment is attached as supplemental file number 1. Choosen miRNAs were as follows (detailed description in Material and Methods section):

Line 195: Mean age of endometriosis patients was 36,6 years, control patients: 58,1 (p<0,01). There was no statistical difference between groups in body mass index (BMI), number of pregnancies, deliveries, miscarriages and comorbidities – detailed characteristics see in suppl 3.

New supplement no 3 was added to the manuscript.

The text was also enriched in new chapter 4.1.

What is the difference between Figure 1 and Figure 2? In Figure 1, miR502 and miR503 levels are roughly in a same range, however, in Figure 2, miR502 level is 1000 times higher? The results are confusing.

Answer: Both figures were corrected. Please see Results section.

The results show some miRNA species are higher in certain types of endometriosis, what do they mean? Can they be linked with pathogenesis of endometriosis?

Minor issue:

In Discussion, the authors wrote “miR125-1-3b is a member of conservative family of proteins”. This is incorrect.

Answer: we corrected this statement – lines 158-159.

Once again we would like to thank all of you for your suggestions and patience during reading and reviewing our study.

On behalf of all co-authors,

Maria Szubert

Round 2

Reviewer 2 Report

 In the revised manuscript, the authors made appropriate corrections to the reviewer’s suggestions. That is, they properly added the descriptions and the figures to the manuscript in accordance with reviewer’s comments.

Author Response

Dear Reviewer, thank you for your acceptance.